# Investigation of the Efficacy of Benzylidene-3-methyl-2-thioxothiazolidin-4-one Analogs with Antioxidant Activities on the Inhibition of Mushroom and Mammal Tyrosinases

**DOI:** 10.3390/molecules29122887

**Published:** 2024-06-18

**Authors:** Hye Jin Kim, Hee Jin Jung, Young Eun Kim, Daeun Jeong, Hyeon Seo Park, Hye Soo Park, Dongwan Kang, Yujin Park, Pusoon Chun, Hae Young Chung, Hyung Ryong Moon

**Affiliations:** 1Department of Manufacturing Pharmacy, College of Pharmacy and Research Institute for Drug Development, Pusan National University, Busan 46241, Republic of Korea; khj3358@pusan.ac.kr (H.J.K.); hjjung2046@pusan.ac.kr (H.J.J.); k87115@pusan.ac.kr (Y.E.K.); 202487128j@pusan.ac.kr (D.J.); gustj6956@pusan.ac.kr (H.S.P.); hyesoo0713@pusan.ac.kr (H.S.P.); 2Department of Medicinal Chemistry, New Drug Development Center, Daegu-Gyeongbuk Medical Innovation Foundation, Daegu 41061, Republic of Korea; kdw4106@kmedihub.re.kr (D.K.); pyj1016@kmedihub.re.kr (Y.P.); 3College of Pharmacy and Inje Institute of Pharmaceutical Sciences and Research, Inje University, Gimhae 50834, Gyeongnam, Republic of Korea; pusoon@inje.ac.kr; 4Department of Pharmacy, College of Pharmacy, Pusan National University, Busan 46241, Republic of Korea; hyjung@pusan.ac.kr

**Keywords:** tyrosinase, melanin, BMTTZD, β-phenyl-α,β-unsaturated carbonyl, docking simulation

## Abstract

Based on the fact that substances with a β-phenyl-α,β-unsaturated carbonyl (PUSC) motif confer strong tyrosinase inhibitory activity, benzylidene-3-methyl-2-thioxothiazolidin-4-one (BMTTZD) analogs **1**–**8** were prepared as potential tyrosinase inhibitors. Four analogs (**1**–**3** and **5**) inhibited mushroom tyrosinase strongly. Especially, analog **3** showed an inhibitory effect that was 220 and 22 times more powerful than kojic acid in the presence of l-tyrosine and l-dopa, respectively. A kinetic study utilizing mushroom tyrosinase showed that analogs **1** and **3** competitively inhibited tyrosinase, whereas analogs **2** and **5** inhibited tyrosinase in a mixed manner. A docking simulation study indicated that analogs **2** and **5** could bind to both the tyrosinase active and allosteric sites with high binding affinities. In cell-based experiments using B16F10 cells, analogs **1**, **3**, and **5** effectively inhibited melanin production; their anti-melanogenic effects were attributed to their ability to inhibit intracellular tyrosinase activity. Moreover, analogs **1**, **3**, and **5** inhibited in situ B16F10 cellular tyrosinase activity. In three antioxidant experiments, analogs **2** and **3** exhibited strong antioxidant efficacy, similar to that of the positive controls. These results suggest that the BMTTZD analogs are promising tyrosinase inhibitors for the treatment of hyperpigmentation-related disorders.

## 1. Introduction

Tyrosinase, a type 3 copper-containing oxidoreductase [1], is a key rate-determining enzyme that regulates melanin content and is widely found in bacteria, plants, animals, and humans [2,3]. It plays key roles in determining skin color, sclerotization of the cuticle in insects, undesirable browning of foods, neurodegeneration, and antibiotic resistance [4,5,6]. In particular, owing to its role in skin pigmentation, it is one of the main targets for pigmented skin disorders such as melasma, vitiligo, lentigines, freckles, age spots, and malignant melanoma in humans [7,8].

Tyrosinase is produced only in melanocytes and plays an essential role in melanin formation [9] due to its critical role as a rate-determining enzyme in melanogenesis. Tyrosinase catalyzes three oxidative reactions in melanogenesis, a series of production processes of the melanin pigments: three conversions of l-tyrosine to l-dopa, l-dopa to dopaquinone, and dihydroxyindole to indole-5,6-quinone [10,11]. Dopaquinone generated by tyrosinase, from l-tyrosine and l-dopa, undergoes several chemicoenzymatic reactions involving enzymes such as tyrosinase-related protein-1 (TRP-1), TRP-2, and tyrosinase to produce eumelanin (black to brown) and pheomelanin (red to yellow). Melanin biosynthesis occurs in melanosomes, organelles of melanocytes present at the basal level of the epidermis [7,12]. Melanin accumulated in melanosomes is transferred to keratinocytes in the epidermis, ultimately determining skin, pupil, and hair color [12]. The oxidation process that converts l-tyrosine to dopaquinone is the rate-determining step in melanin biosynthesis. Therefore, inhibiting tyrosinase activity is the most attractive treatment method for hyperpigmentation. Moreover, clinically used tyrosinase inhibitors including hydroquinone have safety concerns and low efficacy; therefore, it is important to find more effective and safer tyrosinase inhibitors.

Compounds inhibiting tyrosinase can be used as insecticides, food anti-browning agents, or skin lightening materials in industries involving agriculture, food, pharmaceuticals, and cosmetics [13,14]. Recently, many synthetic tyrosinase inhibitors have been shown to exhibit more potent tyrosinase inhibitory activities than natural tyrosinase inhibitors. Compounds that inhibit tyrosinase activity include benzaldehydes [15], thiosemicarbazides [16], stilbenes [17], kojic acids [18], resorcinols [19], cinnamates [20], flavonoids [21,22], curcuminoids [23], azoles [24,25], thioureas [26], and chalcones [27].

Mushroom tyrosinase is located in the cytoplasm and is the most well-characterized tyrosinase, with a tetrameric structure (L_2_H_2_) consisting of two light (L) subunits and two heavy (H) subunits [28]. In addition, mushroom tyrosinase has a high structural similarity to other tyrosinases in the active site. The active site region is well conserved in all tyrosinases, although their origins are different [29]. Mammal tyrosinases, including human tyrosinase, are glycosylated monomers that are anchored to the melanosome membranes of melanocytes [30]. These structural differences may result in different responses to tyrosinase inhibitors. Despite these differences between mushroom and human tyrosinases, mushroom tyrosinase has been widely utilized as a representative enzyme for the discovery of new tyrosinase inhibitors to treat disorders related to hyperpigmentation because it is commercially available, inexpensive, and relatively stable.

Compounds with 3-methyl-2-thioxothiazolidin-4-one (Figure 1) exhibit ATP-noncompetitive GSK-3 [31], plant growth [32], mosquito repellent [33], antibacterial [34], arylamine N-acetyltransferase 1 and 2 inhibitory [35], and antiviral [36] activities. Over the past decades, we identified compounds with a β-phenyl-α,β-unsaturated carbonyl (PUSC) motif (Figure 1) as novel tyrosinase inhibitors [20,37,38]. The PUSC motif confers strong tyrosinase inhibitory ability. In an ongoing effort to discover new tyrosinase inhibitors, attempts were made to synthesize potential tyrosinase inhibitors with the PUSC motif by coupling 3-methyl-2-thioxothiazolidin-4-one with benzaldehyde (Figure 1). Benzylidene-3-methyl-2-thioxothiazolidin-4-one (BMTTZD) analogs were synthesized, and their tyrosinase inhibitory ability was investigated using tyrosinases from mushrooms and B16F10 murine cells. In addition, we examined whether these analogs could inhibit melanogenesis in B16F10 cells. Moreover, their antioxidant efficacy was assessed using 2,2-diphenyl-1-picrylhydrazyl (DPPH) and 2,2′-azino-bis(3-ethylbenzothiazoline-6-sulfonic acid) cation (ABTS^+^) radical scavenging assays and a reactive oxygen species (ROS) scavenging assay. Furthermore, the in situ intracellular tyrosinase activities of the BMTTZD analogs were evaluated using B16F10 cells and l-dopa staining. Although antioxidant activity and anticancer activity are linked [39], we did not conduct research on this because it is outside the topic of this study.

## 2. Results and Discussion

### 2.1. Synthesis of Benzylidene-3-Methyl-2-Thioxothiazolidin-4-One (BMTTZD) Analogs ***1***–***8***

For the synthesis of BMTTZD analogs **1**–**8**, commercially available 3-methyl-2-thioxothiazolidin-4-one was utilized as a key starting component. The target analogs were synthesized using a Knoevenagel condensation reaction as previously described [35]. As illustrated in Figure 1, the reaction of 3-methyl-2-thioxothiazolidin-4-one with appropriately substituted benzaldehydes under Knoevenagel conditions using NaOAc in acetic acid produced eight BMTTZD analogs, i.e., analogs **1**–**8** as the target compounds, with 85–92% yield. The stereochemistry of the trisubstituted double bonds of analogs **1**–**8** was confirmed by ^1^H,^13^C-coupling constants measured in ^1^H-coupled ^13^C NMR mode. Nair et al. reported that the coupling constants between ^1^H and ^13^C can be used to determine the stereochemistry of trisubstituted alkenes [40]. For α,β-unsaturated carbonyl compounds with a trisubstituted double bond, where the carbonyl group and vinyl hydrogen are on the same side, the *J* values range between 3.6 and 7.0 Hz, whereas for α,β-unsaturated carbonyl compounds with a trisubstituted double bond, where the groups are on opposite sides, the *J* values generally exceed 10 Hz. The carbon peak of the carbonyl of analog **3** with a 2,4-dihydroxyphenyl group was recorded as a doublet of quartets in ^1^H-coupled ^13^C NMR mode: a doublet (6.3 Hz) by vinyl hydrogen and a quartet (2.6 Hz) by NCH_3_ (see Appendix A). The corresponding coupling constants of the other analogs were approximately 6.3 Hz. Thus, the C,C double-bond stereochemistry of analogs **1**–**8** was determined to be (*Z*)-forms.

### 2.2. Mushroom Tyrosinase Inhibitory Activities of BMTTZD Analogs ***1***–***8***

The tyrosinase inhibition ability of BMTTZD analogs **1**–**8** was assessed using mushroom tyrosinase. l-Tyrosine or l-dopa and kojic acid were utilized as the substrate and the positive reference substance, respectively. Test samples (BMTTZD analogs and kojic acid) were treated at various concentrations to determine their IC_50_ values. 

All compounds exhibited a concentration-dependent manner for tyrosinase inhibition, regardless of substrate type. As depicted in Table 1, when l-dopa was utilized as a substrate, the IC_50_ value of kojic acid was 24.09 μM. Analog **3** showed the strongest tyrosinase inhibition with an IC_50_ value of 1.12 µM, indicating that analog **3** is a 22-fold stronger tyrosinase inhibitor than kojic acid. The removal of the 2-hydroxyl substituent in analog **3** resulted in a 17-fold reduction in tyrosinase inhibition (analog **1**: IC_50_ value = 17.62 µM), but the tyrosinase inhibitory activity was still potent. Even when the 2-hydroxyl substituent of analog **3** was moved to position 3, the tyrosinase inhibitory activity was reduced (analog **2**: IC_50_ value = 6.18 µM). However, the tyrosinase inhibition of analog **2** was greater than that of analog **1**, and analog **2** still was four times stronger than kojic acid. Analog **4**, which has an additional 3-methoxyl substituent to analog **1**, showed an IC_50_ value >200 μM. The exchange of the methoxyl and hydroxyl substituents of analog **4** greatly raised the tyrosinase inhibition (analog **5**: IC_50_ value = 13.75 µM). Insertion of two methoxyl substituents into the benzene ring of analog **1** significantly decreased its tyrosinase inhibition with an IC_50_ value >200 µM (analog **6**). Introducing one or two bromine substituents on the benzene ring of analog **1** also reduced the tyrosinase inhibition with IC_50_ values >200 µM (analogs **7** and **8**). In summary, analogs **1**–**3** and **5** showed stronger tyrosinase inhibitory activities than kojic acid.

Second, when tested in the presence of l-tyrosine, kojic acid exhibited an IC_50_ value of 17.68 µM. In general, the BMTTZD analogs inhibited tyrosinase more strongly with l-tyrosine than with l-dopa. Analog **1** (IC_50_ value = 3.82 µM) inhibited tyrosinase five times more strongly than kojic acid. The insertion of 3-hydroxyl in analog **1** did not induce a change in tyrosinase inhibition (analog **2**: IC_50_ value = 3.77 µM). Analog **2** (IC_50_ value = 3.77 µM), which has an additional hydroxyl substituent in analog **1**, did not lead to a change in tyrosinase inhibition. However, analog **3** (IC_50_ value = 0.08 µM), which has an additional hydroxyl substituent in analog **1**, dramatically enhanced tyrosinase inhibition, and its inhibitory efficacy was 220 times stronger than that of kojic acid. Analogs **4** (IC_50_ value = 54.81 µM) and **6** (IC_50_ value = 57.40 µM), which have an additional 3-methoxyl or 3,5-dimethoxyl substituent to analog **1**, induced a reduction in tyrosinase inhibition. As observed with l-dopa, the exchange of the substituents of analog **4** resulted in analog **5** with an IC_50_ value of 3.60 µM. Analogs **7** (IC_50_ value = 33.23 µM) and **8** (IC_50_ value = >200 µM), which have an additional one or two bromine substituents to analog **1**, induced a decrease in tyrosinase inhibition. The more bromine was added, the greater the tyrosinase inhibition. Like with l-dopa, analogs **1**–**3** and **5** with l-tyrosine exerted inhibition stronger than kojic acid. However, inhibition of tyrosinase activity in the presence of l-tyrosine was greater than in the presence of l-dopa.

### 2.3. Kinetic Studies of BMTTZD Analogs ***1***–***3*** and ***5*** against Mushroom Tyrosinase

As BMTTZD analogs **1**–**3** and **5** showed potent mushroom tyrosinase inhibitory activities, their tyrosinase inhibitory mechanisms were investigated using Lineweaver–Burk plots. The initial rate of dopachrome generation was examined at various BMTTZD analog concentrations (0, 1.5, 3, and 6 µM for analog **1**; 0, 2, 4, and 8 µM for analogs **2** and **5**; and 0, 0.05, 0.1, and 0.2 µM for **3**) with 0.5, 1, 2, 4, 8, or 16 mM l-dopa as substrates using mushroom tyrosinase. Plotting the reciprocal of the dopachrome formation rate against the inverse of the l-dopa concentration produced Lineweaver–Burk plots (Figure 2). Each Lineweaver–Burk plot created four straight lines with different slopes. For analogs **1** and **3**, the four straight lines met at a point on the y-axis, whereas for analogs **2** and **5**, the four straight lines converged at a point in the second quadrant. These results suggest that for analogs **1** and **3**, the maximal enzymatic rate was constant regardless of the concentration of the inhibitor, indicating that analogs **1** and **3** are competitive inhibitors. For analogs **2** and **5**, the maximal enzymatic rate decreased and the K_M_ value grew as the inhibitor concentration increased, indicating that analogs **2** and **5** are mixed-type inhibitors.

Lineweaver–Burk plots for competitive inhibitors **1** and **3** obtained with l-dopa were converted to corresponding Dixon plots by plotting the reciprocal of the dopachrome formation rate against the inhibitor concentration (Figure 3). The Dixon plots produced lines that met in the second quadrant. The x-coordinate of the convergence position represents the inhibition constant (K_i_) of the ligand. The K_i_ values of analogs **1** and **3** were 6.2 × 10^−6^ and 4.7 × 10^−8^ M, respectively, indicating that analog **3** binds mushroom tyrosinase at the active site much more tightly than analog **1**.

### 2.4. In Silico Docking Simulation Studies of BMTTZD Analogs ***1***–***3*** and ***5*** and Mushroom Tyrosinase at the Active and Allosteric Sites

Because BMTTZD analogs **1**–**3** and **5** strongly inhibited mushroom tyrosinase, a docking simulation between BMTTZD analogs and mushroom tyrosinase was performed using AutoDock Vina to examine their binding affinity and plausible binding interactions. 

The docking results for the ligands (kojic acid (a positive control) and analogs **1**–**3** and **5**) bound to the tyrosinase active site are depicted in Figure 4. Kojic acid interacted with tyrosinase using *pi-pi* stacking and four hydrogen bonds: 2-hydroxymethyl interacting with three histidine residues (His61, His 263, and His296) via hydrogen bonds, 5-hydroxyl interacting with Met280 via a hydrogen bond, and 4-pyranone interacting with His263 via *pi-pi* stacking. Due to these interactions, kojic acid had a binding energy of −5.4 kcal/mol. Three amino acids interacted with analog **1**: the hydroxyl group interacting with Met280 via a hydrogen bond and the phenyl group hydrophobically interacting with Phe264 and Val283, providing a binding energy of −6.5 kcal/mol. Analog **2** interacted with the same amino acids (Phe264, Met280, and Val283) and interaction modes (hydrogen bonding and hydrophobic interactions) as analog **1**, with similar binding energy of −6.5 kcal/mol as analog **1**. Analog **3** created two hydrogen bonds and two hydrophobic interactions: the 2-hydroxyl and 4-hydroxyl groups interacting with Asn260 and Met280, respectively, via a hydrogen bond and the phenyl group hydrophobically interacting with Phe264 and Val283, with −6.8 kcal/mol. Analog **5** produced only a hydrophobic interaction with Val283, with −6.8 kcal/mol. All four analogs showed higher binding affinities to mushroom tyrosinase than kojic acid.

The kinetic results showed that **2** and **5** are mixed-type tyrosinase inhibitors. Therefore, we performed docking simulations to examine whether these analogs can bind to the tyrosinase allosteric site. 

Analogs **2** and **5** were bound to different allosteric sites (Figure 5A,B). The two hydroxyl substituents of analog **2** participated in the formation of two hydrogen bonds (HBs) with Asn22 and His390 (Figure 5C). However, the 4-hydroxyl served as a HB acceptor and the 3-hydroxyl served as a HB donor. In addition, the phenyl ring hydrophobically interacted with Trp386. These interactions provided analog **2** with −5.5 kcal/mol. Analog **5** participated in only one interaction, and the phenyl ring hydrophobically interacted with Thr197, with −5.7 kcal/mol. These results suggest that analogs **2** and **5** can bind to the allosteric site more strongly than kojic acid binds to the active site.

### 2.5. Cytotoxic Effects of Analogs ***1***–***3*** and ***5*** in B16F10 Cells

Because analogs **1**–**3** and **5** exerted potent mushroom tyrosinase inhibitory activities, their effects on anti-melanogenic and cellular tyrosinase activities were evaluated in B16F10 murine cells. Prior to examining their cellular activity, their cell viabilities were investigated. Cell viability was evaluated at 48 and 72 h after treatment with five different concentrations of analogs. 

Analogs **1**, **3**, and **5** did not exhibit cytotoxicity at ≤20 µM at both 48 and 72 h (Figure 6). However, analog **2** exhibited significant cytotoxicity in a concentration-dependent manner, even at 2.5 µM, at both 48 and 72 h. Thus, analog **2** was excluded from experiments on anti-melanogenic and cellular tyrosinase activities in B16F10 cells.

### 2.6. Cellular Tyrosinase Inhibition Effects of Analogs in B16F10 Cells

We investigated whether analogs **1**, **3**, and **5**, which showed potent mushroom tyrosinase inhibitory activities, could inhibit mammalian tyrosinase activity in cell-based experiments. Analogs **1**, **3**, and **5** were exposed to B16F10 cells at various concentrations (20, 10, 5, and 0 µM) before exposure to IBMX (200 µM; 3-isobutyl-1-methylxanthine) and α-MSH (1 µM; α-melanocyte-stimulating hormone). After incubation (72 h), B16F10 cellular tyrosinase activity was determined. As shown in Figure 7, exposure to IBMX plus α-MSH raised tyrosinase activity by 3.9-fold, but exposure to kojic acid (positive control) reduced the tyrosinase activity by 3.6-fold. Analogs **1**, **3**, and **5** concentration-dependently and significantly reduced the stimulators (α-MSH plus IBMX)-induced tyrosinase activity. Analogs **1** and **3** at 10 µM exerted tyrosinase inhibition slightly more strongly than kojic acid. At the highest concentration tested, analogs **1**, **3**, and **5** exhibited stronger inhibitory effects than kojic acid. It was examined whether the cellular tyrosinase inhibition of the BMTTZD analogs could influence B16F10 cellular melanin contents.

### 2.7. Effect of Analogs ***1***, ***3***, and ***5*** on B16F10 Intracellular Melanin Levels

We examined whether the ability of analogs **1**, **3**, and **5** to inhibit intracellular tyrosinase activity could affect intracellular melanin levels. B16F10 cells were utilized as mammalian cells. IBMX and α-MSH were used at 200 µM and 1 µM, respectively, to increase melanin biosynthesis. Kojic acid (positive control) and analogs **1**, **3**, and **5** were exposed to the cells at a concentration of 20 µM and various concentrations (20, 10, 5, and 0 µM). For the intracellular melanin content determination, the absorbance at 405 nm was evaluated after 72 h of incubation. 

As shown in Figure 8, exposure to the stimulators (IBMX and α-MSH) increased melanin levels by 4.4-fold, but exposure to kojic acid decreased the melanin contents by 3.3-fold. Analogs **1**, **3**, and **5** also significantly reduced the melanin contents enhanced by IBMX plus α-MSH. Analogs **1** and **5** at 20 µM reduced the intracellular melanin contents to levels similar to those in treatment with kojic acid. Analog **3** at the highest concentration tested decreased the melanin contents by 2.0-fold. The results for intracellular melanin levels were similar to the results for cellular tyrosinase activity, suggesting that the antimelanogenesis effect of the analogs is probably due to their ability to inhibit tyrosinase.

### 2.8. Evaluation of In Situ Intracellular Tyrosinase Activities of Analogs in B16F10 Cells

Because tyrosinase plays a critical role in melanogenesis, in situ intracellular tyrosinase activity was measured. The in situ intracellular tyrosinase activity was evaluated in B16F10 cells following a previously reported procedure [38]. Stimulators (α-MSH plus IBMX) were exposed to cells to raise tyrosinase activity levels, and analogs **1**, **3**, and **5** were treated at 20, 10, and 5 µM for 72 h. After fixation and permeabilization using paraformaldehyde and Triton X-100, the cells were stained with l-dopa.

Treatment with the stimulators resulted in a great increase in intracellular tyrosinase inhibition (Figure 9). Exposure to kojic acid (20 µM), the positive control, decreased the stimulator-raised intracellular tyrosinase inhibition. Analogs **1**, **3**, and **5** concentration-dependently reduced the stimulator-enhanced intracellular tyrosinase inhibition. Analog **1** showed weaker intracellular tyrosinase inhibition efficacy than kojic acid, but analog **5** greatly exerted intracellular tyrosinase inhibition, similar to kojic acid at the same concentration (20 µM). Moreover, analog **3** at the highest concentration tested exerted tyrosinase inhibition stronger than did kojic acid.

### 2.9. Antioxidant Efficacy of BMTTZD Analogs ***1***–***8***

The antioxidant ability of compounds is reported to be associated with the inhibition of melanin production [41,42]. Thus, the antioxidant ability of BMTTZD analogs **1**–**8** to scavenge radicals of DPPH and ABTS^+^ was evaluated. In addition, the ability of analogs **1**–**8** to scavenge in vitro ROS was assessed.

First, the ability of BMTTZD analogs **1**–**8** to scavenge DPPH radicals was examined using l-ascorbic acid (vitamin C; positive control). All samples were tested at 500 µM. 

l-Ascorbic acid scavenged DPPH radicals by 97% (Figure 10A). The BMTTZD analogs exhibited radical scavenging activities ranging from very weak to high. The more hydroxyl groups the analogs had, the greater the DPPH radical scavenging capacity (analogs **2** and **3** vs. analogs **1** and **4**–**8**). Analog **2** and analog **3** scavenged 93% and 82% of DPPH radicals, respectively. The ability of analog **2** to remove DPPH radicals was similar to that of the positive control, vitamin C. Analogs (**1** and **5**–**8**) with only one OH group on the benzene ring exerted a low ability to remove DPPH radicals (7–30% inhibition). On the other hand, analog **4** had a strong ability to remove DPPH radicals (82% scavenging activity).

Second, the ability of BMTTZD analogs to scavenge ABTS^+^ radicals was examined. ABTS^+^ radicals were obtained via the oxidation of ABTS using potassium persulfate. All samples (Trolox (positive control) and **1**–**8**) were tested at 100 µM. 

Trolox treatment scavenged the ABTS^+^ radicals by 95% (Figure 10B). As in the DPPH radical scavenging experiment, analogs **2** and **3** having more hydroxyl groups exhibited greater activities to scavenge ABTS^+^ radicals, with 92% and 95% scavenging activity, respectively. Analogs **4**–**6**, with 3-methoxy-4-hydroxyl, 4-hydroxy-3-methyl, and 3,5-dimethoxy-4-hydroxyl groups, showed moderate activity, with 57–72% radical scavenging activity. The remaining analogs exhibited less than 30% radical scavenging activity.

Third, the ability of analogs **1**–**8** to scavenge in vitro ROS was determined. Esterase was mixed with 2′,7′-dichlorodihydrofluorescein diacetate (DCFH-DA) to produce 2′,7′-dichlorodihydrofluorescein (DCFH). 3-Morpholinosydnonimine (SIN-1) generates ROS that react with DCFH, resulting in the formation of DCF (2′,7′-dichlorofluorescein). Therefore, the ability of analogs **1**–**8** to scavenge ROS was calculated by measuring the fluorescence of DCF with 10 µM SIN-1 and 40 µM test samples (Trolox (positive control) and analogs **1**–**8**). Figure 10C shows the results of ROS scavenging activity of each sample.

SIN-1 treatment greatly raised ROS levels, and exposure to Trolox significantly decreased the SIN-1-induced ROS contents. Analogs **4**–**8** did not significantly alter or only slightly altered the ROS levels induced by SIN-1. However, analogs **1**–**3**, which have 4-hydroxyphenyl, catechol, and resorcinol groups, respectively, reduced the SIN-1-induced ROS levels to the same extent as Trolox. In summary, analogs **2** and **3** exhibited strong antioxidant effects.

## 3. Materials and Methods

### 3.1. Chemistry

#### 3.1.1. General Methods

Chemical reagents and solvents were acquired from DaeJung (Siheung-si, Gyeonggi, Republic of Korea), Sigma Co. (St. Louis, MO, USA), Thermo Fisher Scientific (Waltham, MA, USA), and SEJIN CI Co. (Seoul, Republic of Korea). All the reagents and solvents were used without purification. TLC (Silica gel 60F_254_, Merck Millipore, Darmstadt, Germany) was utilized to monitor the reaction progress. ^1^H-NMR spectra of all products were acquired utilizing a Varian Unity AS500 unit spectrometer (Agilent Technologies, Santa Clara, CA, USA), and ^13^C-NMR spectra of all products were acquired using a JEOL ECZ400S (JEOL Ltd., Tokyo, Japan) and a Varian Unity AS500 unit spectrometer. Mass data were acquired on a High Resolution Liquid Chromatography tandem Mass Spectrometer (Sciex’s Zeno TOF 7600 mass instrument system; Toronto, ON, CAN). DMSO-*d*_6_ was utilized as the NMR solvent. The *J* and *δ* values of NMR data were presented in Hz and ppm, respectively. The splitting patterns of NMR peaks are as follows: m (multiplet), d (doublet), s (singlet), brs (broad singlet), dq (doublet of quartets), and dd (doublet of doublets).

#### 3.1.2. Procedure for the Synthesis of BMTTZD Analogs **1**–**8**

A solution of 3-methylrhodanine (200 mg, 1.36 mmol), benzaldehyde (1.1 equiv. for benzaldehydes with no bromo group and 0.9 equiv. for benzaldehydes with bromo groups), and NaOAc (3.0 equiv.) in acetic acid (1.3 mL) was heated at reflux for 3–14 h. After cooling, cold water was added to obtain a precipitate. The precipitate was filtered and washed with water, dichloromethane, methanol, and hexane:dichloromethane (1:1) depending on the remaining starting material. The filter cake was obtained as a pure compound in 85–92% yield. For ^1^H and ^13^C NMR spectra, see Appendix A.

*(Z)-5-(4-Hydroxybenzylidene)-3-methyl-2-thioxothiazolidin-4-one* (analog **1**)

4 h; 92%; ^1^H NMR (DMSO-*d*_6_, 500 MHz) *δ* 10.46 (brs, 1H), 7.68 (s, 1H), 7.47 (d, *J* = 8.5 Hz, 2H), 6.91 (d, *J* = 8.5 Hz, 2H), 3.36 (s, 3H); ^13^C NMR (DMSO-*d*_6_, 125 MHz) *δ* 193.8, 167.5, 161.0, 134.0, 133.7, 124.4, 118.4, 117.0, 31.5; HRMS (EDA) *m*/*z* C_11_H_10_NO_2_S_2_ (M + H)^+^ calcd 252.0147, obsd 252.0148.

*(Z)-5-(3,4-Dihydroxybenzylidene)-3-methyl-2-thioxothiazolidin-4-one* (analog **2**)

9 h; 92%; ^1^H NMR (DMSO-*d*_6_, 500 MHz) *δ* 9.75 (brs, 2H), 7.58 (s, 1H), 7.00 (s, 1H), 6.99 (d, *J* = 8.0 Hz, 1H), 6.86 (d, *J* = 8.0 Hz, 1H), 3.34 (s, 3H); ^13^C NMR (DMSO-*d*_6_, 125 MHz) *δ* 193.8, 167.5, 149.9, 146.5, 134.4, 125.7, 124.9, 118.2, 117.2, 116.9, 31.5.

*(Z)-5-(2,4-Dihydroxybenzylidene)-3-methyl-2-thioxothiazolidin-4-one* (analog **3**)

9 h; 86%; ^1^H NMR (DMSO-*d*_6_, 500 MHz) *δ* 10.68 (s, 1H), 10.35 (s, 1H), 7.94 (s, 1H), 7.18 (d, *J* = 8.5 Hz, 1H), 6.44–6.39 (m, 2H), 3.36 (s, 3H); ^13^C NMR (DMSO-*d*_6_, 125 MHz) *δ* 194.0, 167.7 (dq, *J* = 5.8, 2.0 Hz), 163.1, 160.4, 131.9, 129.7, 116.2, 112.4, *109*.4, 102.9, 31.5; HRMS (EDA) *m*/*z* C_11_H_10_NO_3_S_2_ (M + H)^+^ calcd 268.0097, obsd 268.0096.

*(Z)-5-(4-Hydroxy-3-methoxybenzylidene)-3-methyl-2-thioxothiazolidin-4-one* (analog **4**)

5 h; 89%; ^1^H NMR (DMSO-*d*_6_, 500 MHz) *δ* 10.14 (brs, 1H, OH), 7.74 (s, 1H, vinylic H), 7.20 (s, 1H), 7.13 (d, *J* = 7.5 Hz, 1H), 6.95 (d, *J* = 7.5 Hz, 1H), 3.84 (s, 3H), 3.39 (s, 3H); ^13^C NMR (DMSO-*d*_6_, 100 MHz) *δ* 193.8, 167.5, 151.0, 148.9, 134.4, 125.9, 125.2, 118.9, 117.1, 115.5, 56.5, 31.6.

*(Z)-5-(3-Hydroxy-4-methoxybenzylidene)-3-methyl-2-thioxothiazolidin-4-one* (analog **5**)

14 h; 90%; ^1^H NMR (DMSO-*d*_6_, 500 MHz) *δ* 9.53 (s, 1H), 7.58 (s, 1H), 7.07 (dd, *J* = 8.5, 2.0 Hz, 1H), 7.02 (d, *J* = 8.5 Hz, 1H), 6.99 (d, *J* = 2.0 Hz, 1H), 3.82 (s, 3H), 3.33 (s, 3H); ^13^C NMR (DMSO-*d*_6_, 125 MHz) *δ* 193.8, 167.4, 151.1, 147.5, 133.9, 126.1, 125.1, 119.4, 116.6, 112.9, 56.2, 31.5; HRMS (EDA) *m*/*z* C_12_H_12_NO_3_S_2_ (M + H)^+^ calcd 282.0253, obsd 282.0257.

*(Z)-5-(4-Hydroxy-3,5-dimethoxybenzylidene)-3-methyl-2-thioxothiazolidin-4-one* (analog **6**)

6 h; 91%; ^1^H *NMR* (DMSO-*d*_6_, 500 MHz) *δ* 9.51 (brs, 1H), 7.66 (s, 1H), 6.86 (s, 2H), 3.81 (s, 6H), 3.33 (s, 3H); ^13^C NMR (DMSO-*d*_6_, 125 MHz) *δ* 193.5, 167.4, 148.7, 139.9, 134.6, 123.7, 118.8, 109.1, 56.5, 31.5.

*(Z)-5-(3-Bromo-4-hydroxybenzylidene)-3-methyl-2-thioxothiazolidin-4-one* (analog **7**)

5 h; 85%; ^1^H NMR (DMSO-*d*_6_, 500 MHz) *δ* 11.28 (brs, 1H), 7.81 (d, *J* = 2.0 Hz, 1H), 7.68 (s, 1H), 7.44 (dd, *J* = 8.5, 2.0 Hz, 1H), 7.09 (d, *J* = 8.5 Hz, 1H), 3.36 (s, 3H); ^13^C NMR (DMSO-*d*_6_, 100 MHz) *δ* 193.6, 167.4, 157.4, 136.8, 132.4, 131.7, 126.1, 120.2, 117.6, 110.9, 31.7.

*(Z)-5-(3,5-Dibromo-4-hydroxybenzylidene)-3-methyl-2-thioxothiazolidin-4-one* (analog **8**)

3 h; 89%; ^1^H NMR (*DMSO*-*d*_6_, 500 MHz) *δ* 10.91 (brs, 1H), 7.75 (s, 2H), 7.66 (s, 1H), 3.36 (s, 3H); ^13^C NMR (DMSO-*d*_6_, 100 MHz) *δ* 193.3, 167.3, 153.8, 134.9, 130.7, 127.8, 122.3, 113.0, 31.7.

### 3.2. Mushroom Tyrosinase Inhibition Assay

The ability of the BMTTZD analogs to inhibit mushroom tyrosinase was evaluated in the presence of l-dopa and l-tyrosine as previously described [38]. An aliquot (20 µL) of the aqueous mushroom tyrosinase solution (500 units/mL for l-dopa and 1000 units/mL for l-tyrosine) was mixed with BMTTZD analogs (10 µL; final concentrations: 50, 10, 2, 0.4, 0.08 and 0.016 µM) or kojic acid (10 µL; final concentrations: 50, 10, and 2 µM) and a substrate mixture (170 µL) consisting of 17.2 mM sodium phosphate buffer (pH 6.5) and 345 µM of l-dopa or l-tyrosine in each well of a 96-well plate. Dopachrome amounts generated during incubation (30 min at 37 °C) were calculated by measuring the well absorbance at 475 nm using a VersaMax^®^ reader (Molecular Devices, Sunnyvale, CA, USA). 

### 3.3. The Kinetic Study of Mushroom Tyrosinase Inhibition of BMTTZD Analogs 

Lineweaver–Burk (L-B) plots were acquired for the kinetic analysis of analogs against mushroom tyrosinase as previously described [37]. In brief, 10 µL of a solution of DMSO and BMTTZD analogs **1**–**3** or **5** (final concentrations: 0, 1.5, 3, and 6 µM for analog **1**; 0, 2, 4, and 8 µM for analogs **2** and **5**; and 0, 0.05, 0.1, and 0.2 µM for analog **3**) in each well of a 96-well plate was mixed with 20 µL of aqueous mushroom tyrosinase solution (200 units/mL) and 170 µL of an aqueous substrate mixture (various concentrations of l-dopa (final concentrations: 16, 8, 4, 2, 1 and 0.5 mM) and 17.2 mM sodium phosphate buffer (pH 6.5)). During incubation (15 min at 37 °C), the increase in the well absorbance (ΔOD_475_/min) was calculated at 1 min intervals utilizing a VersaMax^®^ reader. The L-B plots were acquired by plotting the inverse of the substrate concentrations against the inverse of the optical density increase at 475 nm per min (ΔOD_475_/min) with a variety of concentrations of BMTTZD analogs. 

### 3.4. In Silico Docking Simulation of Mushroom Tyrosinase and BMTTZD Analogs

Virtual docking simulations between tyrosinase and BMTTZD analogs were carried out utilizing AutoDock Vina docking software (ver. 1.1.2), as described previously [20]. For the docking simulations, the mushroom (*A. bisporus*) tyrosinase X-ray structure (PDB ID: 2Y9X) was utilized. The 2D structures of the ligands (kojic acid (positive substance) and BMTTZD analogs) were acquired utilizing ChemDraw Ultra and transformed into the corresponding 3D structures utilizing Chem3D Pro 12.0. The binding energies between the BMTTZD ligands and tyrosinase were obtained using Chimera 35.91.958 and AutoDock Vina. Pharmacophores exhibiting the possible interactions between the tyrosinase amino acid residues and the ligands were obtained utilizing LigandScout 4.4.

### 3.5. B16F10 Cell Culture

B16F10 cells were acquired from American Type Culture Collection (Manassas, VA, USA). These cells were cultivated in DMEM supplemented with 100 µg/mL streptomycin, FBS, and 100 IU/mL penicillin at 37 °C with 5% CO_2_.

### 3.6. Cell Cytotoxicity Test

The effect on B16F10 cytotoxicity was assessed following a previously described method [22]. Cells (1 × 10^4^ cells/well of a 96-well plate) were cultivated for 24 h with 5% CO_2_ at 37 °C. The cells were exposed to the BMTTZD analogs (test concentrations: 20, 10, 5, 2.5, and 0 µM) and cultivated with 5% CO_2_ at 37 °C for 72 h. An EZ-Cytox solution (10 µL; EZ-500, DoGenBio, Seoul, Republic of Korea) was exposed to each well, and the plate was cultivated for 2 h. Cell viability was calculated from the well absorbance measured at 450 nm utilizing a reader (VersaMax^®^).

### 3.7. Cellular Tyrosinase Activity Assay

Effects on B16F10 tyrosinase inhibition were evaluated as previously described [38]. A 6-well plate containing 1 × 10^5^ B16F10 cells/well was cultivated for 23 h under conditions identical to those of the cell culture. The cultivated cells were exposed to 20 µM kojic acid (positive substance) or analogs **1**–**3** and **5** (final concentration: 20, 10, 5, and 0 µM) for 1 h, followed by 200 µM IBMX and 1 µM α-MSH. The cells were cultivated for 72 h. After washing 2 times with PBS, the cells were lysed in a 100 µL lysis buffer, including phenylmethylsulfonyl fluoride (5 µL; 1 mM), Triton X-100 (5 µL; 1%), and phosphate buffer (90 µL; 50 mM (pH 6.5)). After 30 min incubation at −80 °C, the cell lysates were centrifuged at 4 °C for 30 min. The supernatants (80 µL) of the lysates were moved to each well of a microplate and mixed with 20 µL 10 mM l-dopa. The well absorbance at 475 nm was recorded for 1 h at 10 min intervals by utilizing a plate reader.

### 3.8. Melanin Content Assay in B16F10 Cells

The effect of BMTTZD analogs on melanin biosynthesis was investigated following a previously described method [38]. A 6-well microplate containing 1 × 10^5^ B16F10 cells/well was cultivated for 23 h under conditions identical to those of the cell culture. The cultivated cells were treated with **1**–**3** and **5** (final concentration: 20, 10, 5, and 0 µM) or kojic acid (20 µM; positive control) for 1 h before exposure to IBMX (200 µM) and α-MSH (1 µM). After exposure to IBMX and α-MSH, B16F10 cells were cultivated for 72 h at 37 °C. The cells were rinsed 2 times with PBS, treated with 1N-NaOH aqueous solution (100 µL), and cultivated at 60 °C for 1 h. The lysates were moved to a microplate. At 405 nm, the well absorbance was measured utilizing a VersaMax^®^ reader. Each experiment was carried out three times independently.

### 3.9. DPPH Radical Scavenging Activity Assay

The ability of the BMTTZD analogs to scavenge DPPH radicals was evaluated as previously reported [22]. To each well of a 96-well plate containing analogs **1**–**8** (5 mM in dimethyl sulfide; 20 µL) was added a DPPH solution (0.2 mM in methanol; 180 µL). The mixture was kept in a place without light for 30 min at 25°C to generate DPPH radicals. The well absorbance at 517 nm was measured utilizing a VersaMax^®^ reader. The ability of the analogs to remove DPPH radicals was compared to that of vitamin C.

### 3.10. ABTS^+^ Radical Scavenging Activity Assay

The ability of the BMTTZD analogs to scavenge ABTS^+^ radicals was assessed as previously described [22], with minor modifications. To a 50 mL tube were added 20 mL aqueous K_2_S_2_O_8_ solution (2.45 mM) and 20 mL aqueous ABTS solution (7 mM). The combined mixture was placed at 22 °C in a place without light for 17 h to generate the ABTS^+^ radicals. For adequate absorbance control (0.70 ± 0.02 at 734 nm), methanol was added to the combined mixture. Test compounds (Trolox and analogs **1**–**8**; 10 µL in 10% DMSO/90% EtOH [*v*/*v*] solution) were mixed with 90 µL ABTS^+^ radical solution. After storage in a place without light for 2 min, the well absorbance at 734 nm was acquired for 10 min every 1 min by using a microplate reader (VersaMax^®^). All test compounds were tested at 100 µM. The activity of the analogs was compared to that of Trolox.

### 3.11. ROS Scavenging Activity Assay

The ability of analogs to scavenge ROS was examined as previously described [37]. To a 15 mL tube were added DCFH-DA solution (1.25 mM; 50 µL), phosphate buffer (50 mM, 4.9 mL), and esterase (600 units/mL; 50 µL). The esterase–DCFH-DA solution was left in a dark place for 30 min. A 10 µL SIN-1 solution was added to each well of a black 96-well plate and mixed with phosphate buffer (180 µL) and Trolox (10 µL) or analogs **1**–**8** (10 µL). SIN-1, Trolox, and analogs **1**–**8** were treated at 10, 40, and 40 µM, respectively. After 5 min, 50 µL of the esterase–DCFH-DA solution was added to each well. The well fluorescence of DCF was acquired at 535 nm utilizing a microplate reader (Berthold Advances GmbH & Co., Bad Wildbad, Germany) using a 485 nm excitation wavelength. Trolox was used to compare ROS scavenging activity.

### 3.12. In Vitro and In Situ Cellular Tyrosinase Activity Assay Using l-Dopa Staining

The in situ cellular tyrosinase activity of BMTTZD analogs **1**, **3**, and **5** was assayed utilizing B16F10 cells and kojic acid (a positive control), as previously described [43]. A 24-well plate containing 1 × 10^3^ B16F10 cells/well was cultivated for 24 h under conditions identical to those of the cell culture. The cells were exposed to 20 µM kojic acid or analogs **1**, **3**, and **5** (final concentration: 20, 10, and 5 µM) for 1 h, followed by exposure to 200 µM IBMX and 1 µM α-MSH to enhance enzyme activity. The cells were then cultivated under conditions identical to those of the cell culture for 72 h. Cultivated cells were fixed for 40 min with paraformaldehyde (4%), washed 2 times with PBS, and permeabilized with Triton X-100 (0.1%) for 2 min. After washing 2 times with PBS, the cells in the 24-well plate were stained with a 2 mM l-dopa aqueous solution (500 µL/well) for 2 h at 37 °C. Staining was photographed and analyzed utilizing a camera connected to a microscope (Motic, Hong Kong).

### 3.13. Statistical Analysis

Based on the three separate experiments, results are presented as the mean ± SEM. The significance was determined by one-way ANOVA followed by the Bonferroni post-hoc test utilizing GraphPad Prism 5 software (La Jolla, CA, USA). *p* < 0.05 was considered significant.

## 4. Conclusions

BMTTZD analogs with the PUSC motif were synthesized under Knoevenagel condensation conditions. Analogs **1**–**3** and **5** exerted stronger mushroom tyrosinase inhibition than kojic acid. Analog **3** was a tyrosinase inhibitor that is 22 and 220 times stronger than kojic acid in the presence of l-dopa and l-tyrosine, respectively. The mechanism of **1**–**3** and **5** was determined through kinetic studies, which was supported by the docking results. Analogs **1**, **3**, and **5** decreased intracellular melanin levels by inhibiting tyrosinase activity. The in situ tyrosinase activity results suggest that analogs **1**, **3**, and **5** inhibit melanogenesis via tyrosinase inhibition in B16F10 cells.

## Data Availability

The data presented in this study are available in article and Appendix A.

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
