# Peer review of "Investigation of the Efficacy of Benzylidene-3-methyl-2-thioxothiazolidin-4-one Analogs with Antioxidant Activities on the Inhibition of Mushroom and Mammal Tyrosinases"

_molecules, 2024, doi:10.3390/molecules29122887_

Round 1

Reviewer 1 Report

Comments and Suggestions for Authors

the authors present medicinal chemistry work for antioxidant activities. the rationale leading to the design of the compounds is relevant and the methodology for accessing analogues is extremely simple. on the other hand, the study of biological activities is quite complete, including anti-oxidant power, enzymatic inhibition and biological activities. In particular, an effort to link antioxidant activities and anti-tumor activities should be highlighted.

Few modifications requested: the minor ones are in the attached pdf (italic, bold..). However, it would be wise:

-add the values of the constants in the experimental part, as well as the HRMS of the compounds. -purity: NMR spectra show the purity of the synthesized compounds, which validates biological studies. However, the purity could be checked by chromatography.

-specify the field of the NMR spectrometer

-figures 4, 7 and 8 are not clear

Author Response

the authors present medicinal chemistry work for antioxidant activities. the rationale leading to the design of the compounds is relevant and the methodology for accessing analogues is extremely simple. on the other hand, the study of biological activities is quite complete, including anti-oxidant power, enzymatic inhibition and biological activities. In particular, an effort to link antioxidant activities and anti-tumor activities should be highlighted.

Response: Thank you for your valuable comment. We aimed to find compounds that inhibit melanogenesis. Although the reviewer provided valuable opinion (efforts to link antioxidant and anticancer activity), it is outside the purpose of this paper. Please understand that we did not do the research suggested by the reviewer. Thus, we have added ‘Although antioxidant activity and anticancer activity are linked [reference 39], we did not conduct research on this because it is outside the topic of this study.’ in the text.

Few modifications requested: the minor ones are in the attached pdf (italic, bold..). However, it would be wise:

Response: Thank you for your valuable suggestion. As suggested by the reviewer, the minor ones (italic, bold, etc.) were revised.

-add the values of the constants in the experimental part, as well as the HRMS of the compounds. -purity: NMR spectra show the purity of the synthesized compounds, which validates biological studies. However, the purity could be checked by chromatography.

Response: Thank you for your valuable suggestion. As suggested by the reviewer, the HRMS data and spectra of the compounds, which were subjected to biological testing using B16F10 cells, have been added in the manuscript and Supplementary Information, respectively. So, it can be confirmed from HRMS and NMR spectra that the purity is sufficient to evaluate biological activity.

-specify the field of the NMR spectrometer

Response: We have already indicated the type (model) of NMR spectrometer and the MHz of the instrument used in the General Method section.

-figures 4, 7 and 8 are not clear

Response: Thank you for your valuable suggestion. We have improved the quality of figures 4, 7, and 8.

Reviewer 2 Report

Comments and Suggestions for Authors

Present manuscript deals with the extension of various cinnamate derivatives with 3-Me-2-thioxothiazolidin-4-4one as a potential tyrosinase inhibitors. This research is a continuation of previous findings reported by the same group. Total 8-new compounds were synthesized, characterized and tested for their antioxidant activities on the mushroom and mammal tyrosinases inhibition.

1.       Author should clarify the difference in the activities of compounds depicated in Ref 20, 37 and 38.

2.       Especially, ref-38 which discloses the thiazol-4(5H)-one derivatives which is similar to the 3-methyl-2-thioxothiazolidin-4-one based compounds reported in the present manuscript. Provide the explanation for the synthesizing these types of analogs.

3.       If tyrosinase activity is based on the cinnamate backbone, what is the reason for hybridizing it with various heterocycles reported in this ref 37, 38 and present manuscript. Overall, it seems there is no effect of this backbone.

4.       Based on the docking study, it seems that there are no additional benefits of synthesizing these compounds as it does not provide any additional benefits in terms of binding interactions when compared with compounds reported in reference 38.

5.       Scheme 1: Provide UPAC names for benzaldehydes a-h instead of structures.

Comments on the Quality of English Language

Present manuscript deals with the extension of various cinnamate derivatives with 3-Me-2-thioxothiazolidin-4-4one as a potential tyrosinase inhibitors. This research is a continuation of previous findings reported by the same group. Total 8-new compounds were synthesized, characterized and tested for their antioxidant activities on the mushroom and mammal tyrosinases inhibition.

1.       Author should clarify the difference in the activities of compounds depicated in Ref 20, 37 and 38.

2.       Especially, ref-38 which discloses the thiazol-4(5H)-one derivatives which is similar to the 3-methyl-2-thioxothiazolidin-4-one based compounds reported in the present manuscript. Provide the explanation for the synthesizing these types of analogs.

3.       If tyrosinase activity is based on the cinnamate backbone, what is the reason for hybridizing it with various heterocycles reported in this ref 37, 38 and present manuscript. Overall, it seems there is no effect of this backbone.

4.       Based on the docking study, it seems that there are no additional benefits of synthesizing these compounds as it does not provide any additional benefits in terms of binding interactions when compared with compounds reported in reference 38.

5.       Scheme 1: Provide UPAC names for benzaldehydes a-h instead of structures.

Author Response

Present manuscript deals with the extension of various cinnamate derivatives with 3-Me-2-thioxothiazolidin-4-4one as a potential tyrosinase inhibitors. This research is a continuation of previous findings reported by the same group. Total 8-new compounds were synthesized, characterized and tested for their antioxidant activities on the mushroom and mammal tyrosinases inhibition.

  1. Author should clarify the difference in the activities of compounds depicated in Ref 20, 37 and 38.
  2. Especially, ref-38 which discloses the thiazol-4(5H)-one derivatives which is similar to the 3-methyl-2-thioxothiazolidin-4-one based compounds reported in the present manuscript. Provide the explanation for the synthesizing these types of analogs.
  3. If tyrosinase activity is based on the cinnamate backbone, what is the reason for hybridizing it with various heterocycles reported in this ref 37, 38 and present manuscript. Overall, it seems there is no effect of this backbone.

Response to comments 1, 2, and 3: Thank you for your valuable comment. The reasons for hybridizing various heterocycles with the cinnamate backbone (β-phenyl-α,β-unsaturated carbonyl [PUSC]) are as follows. In the cinnamate backbone with an open chain, the angle between the carbonyl and the single bond and the angle between the single and double bonds are 120 degrees, but when hybridized with heterocycles, these angles change somewhat, which may affect the tyrosinase inhibitory activity. The IC50 values of the compounds with the 2,4-dihydroxyphenyl moiety in references 37 and 38 are 5 and 0.4 uM, respectively, and the corresponding compound 3 in this manuscript has an IC50 value of 0.08 uM, showing 63 and 5 times more potent mushroom tyrosinase inhibitory activity than the two compounds in references 37 and 38. These results may be due to differences in bond angles.

  1. Based on the docking study, it seems that there are no additional benefits of synthesizing these compounds as it does not provide any additional benefits in terms of binding interactions when compared with compounds reported in reference 38.

Response: Thank you for your valuable comment. Docking studies play a very useful role in finding compounds with excellent activity. However, docking results do not always match biological results. Additionally, there are many cases where different results are obtained depending on the docking program when docking the same compound. Therefore, I think we should be very cautious in drawing conclusions based solely on the docking results. In the case of the compounds in this manuscript, the carbonyl group and double bond do not interact in the docking results. So, we have previously performed tyrosinase inhibitory activity tests on compounds with these parts (the carbonyl and the double bond) deleted. As a result, we have confirmed that there is almost no tyrosinase inhibitory activity. Therefore, we believe that even parts that do not interact directly in the docking results can sensitively affect activity.

  1. Scheme 1: Provide UPAC names for benzaldehydes a-h instead of structures.

Response: Thank you for your valuable comment. As suggested by the reviewer, we have deleted the structures of benzaldehydes and provided IUPAC names for benzaldehydes a-h.

Reviewer 3 Report

Comments and Suggestions for Authors

This manuscript describes tyrosinase inhibition activity of benzylidene-3-methyl-2-thioxothiazolidine-4-on analogues including the kinetic studies and docking simulation studies.  The synthesized compounds showed more potent activity than positive control.  In addition, there are many interesting results in this paper.  Therefore, I think this manuscript could be acceptable as an original paper in Molecules after considering the following comments.

I am not sure the reason why the authors chose benzylidene-3-methyl-2-thioxothiazolidin-4-one in this study.  I hope to be added the clear reason, if possible.

I think the hydrophobic interaction have little influence.  I hope to explain the effects of the interaction, if possible.

Author Response

This manuscript describes tyrosinase inhibition activity of benzylidene-3-methyl-2-thioxothiazolidine-4-on analogues including the kinetic studies and docking simulation studies.  The synthesized compounds showed more potent activity than positive control.  In addition, there are many interesting results in this paper.  Therefore, I think this manuscript could be acceptable as an original paper in Molecules after considering the following comments.

I am not sure the reason why the authors chose benzylidene-3-methyl-2-thioxothiazolidin-4-one in this study.  I hope to be added the clear reason, if possible.

Response: Thank you for your valuable comment. Because 3-methyl-2-thioxothiazolidin-4-one was commercially available, the compound was selected among several heterocycles. The reasons for hybridizing the compound with the β-phenyl-α,β-unsaturated carbonyl [PUSC] are as follows. For compounds with a β-phenyl-α,β-unsaturated carbonyl [PUSC] scaffold in open chain form, the angle between the carbonyl and the single bond and the angle between the single and double bonds are 120 degrees. However, when the PUSC scaffold is hybridized with a heterocycle such as 3-methyl-2-thioxothiazolidin-4-one, these angles change somewhat, which may affect the tyrosinase inhibitory activity. Please refer to references 20, 37, and 38.

I think the hydrophobic interaction have little influence.  I hope to explain the effects of the interaction, if possible.

Response: In general, hydrophobic interactions are weaker than hydrogen bonds. However, the degree of interaction (binding energy) of hydrogen bonding also varies depending on the bond length and bond angle, and in hydrophobic interaction, if two interacting groups are located close to each other, it is possible to interact with each other more strongly than hydrogen bonding.

Reviewer 4 Report

Comments and Suggestions for Authors

The manuscript by Kim et al. reports the characterization of some benzylidene-3-methyl-2-thioxo- 2-thiazolidin-4-one (BMTTZD) analogs as inhibitors of tyrosinase, an enzyme involved in hyperpigmentation-related disorders. The multidisciplinary approach involving enzymology, molecular modeling and cell-based methodologies allowed the identification of three BMTTZD analogs inhibiting cellular tyrosinase and melanogenesis in B16F10 cells, as promising lead compounds.

The data are well presented and described in terms of details. However, in Figures 2 and 3 the standard errors in the activity points are missing. If the data presented correspond to one of three “independently conducted” experiments, at least this should be mentioned.

Comments on the Quality of English Language

The English language needs only minor style editing, as in some parts of the manuscript it looks like spoken rather than written English.

Author Response

The manuscript by Kim et al. reports the characterization of some benzylidene-3-methyl-2-thioxo- 2-thiazolidin-4-one (BMTTZD) analogs as inhibitors of tyrosinase, an enzyme involved in hyperpigmentation-related disorders. The multidisciplinary approach involving enzymology, molecular modeling and cell-based methodologies allowed the identification of three BMTTZD analogs inhibiting cellular tyrosinase and melanogenesis in B16F10 cells, as promising lead compounds.

The data are well presented and described in terms of details. However, in Figures 2 and 3 the standard errors in the activity points are missing. If the data presented correspond to one of three “independently conducted” experiments, at least this should be mentioned.

Response: Thank you for your valuable comment. As suggested by the reviewer, we have added ‘One of the data independently conducted is presented’ in the legend of Figures 2 and 3.